# Unraveling Allelic Impacts on Pre-Harvest Sprouting Resistance in *TaVP1*-B of Chinese Wheat Accessions Using Pan-Genome

**DOI:** 10.3390/plants14040504

**Published:** 2025-02-07

**Authors:** Danfeng Wang, Jinjin Xie, Jingwen Wang, Mengdi Mu, Haifeng Xiong, Fengshuo Ma, Peizhen Li, Menghan Jia, Shuangjing Li, Jiaxin Li, Mingyue Zhu, Peiwen Li, Haiyan Guan, Yi Zhang, Hao Li

**Affiliations:** 1Key Laboratory of Plant Genetics and Molecular Breeding, Zhoukou Normal University/Henan Key Laboratory of Crop Molecular Breeding & Bioreactor, Zhoukou 466001, China; wdf19911025@live.com (D.W.); 13095690517@163.com (J.W.); 17203791658@163.com (M.M.); 2State Key Laboratory of Crop Stress Adaptation and Improvement, College of Agriculture, Henan University, Kaifeng 475000, China; xiejinjin6666@163.com; 3Henan International Joint Laboratory of Translational Biology, Zhoukou 466001, China; hyguan0820@163.com; 4Henan Plant Gene and Molecular Breeding Engineering Research Center, Zhoukou 466001, China; 17839721223@163.com; 5Henan Province Plant Genetics and Molecular Breeding Innovation Team, Zhoukou 466001, China; 15670410885@163.com (F.M.); 13569272016@163.com (P.L.); jia200512120186@163.com (M.J.); 13723028501@163.com (S.L.); 18836733512@163.com (J.L.); 19712600307@163.com (M.Z.); blingbulingbiu@163.com (P.L.); 6Henan Crop Molecular Design Breeding and Cultivation Engineering Technology Research Center, Zhoukou 466001, China

**Keywords:** wheat, pre-harvest sprouting, *TaVP1*-B, haplotype, KASP

## Abstract

The *TaVP1-B* gene, located on the 3B chromosome of wheat, is a homolog of the *Viviparous-1* (*VP-1*) gene of maize and was reported to confer resistance to pre-harvest sprouting (PHS) in wheat. In this study, the structure of the *TaVP1-B* gene was analyzed using the wheat pan-genome consisting of 20 released cultivars (19 wheat are from China), and 3 single nucleotide polymorphisms (SNPs), which were identified at the 496 bp, 524 bp, and 1548 bp of the *TaVP1-B* CDS region, respectively. Haplotypes analysis showed that these SNPs were in complete linkage disequilibrium and that only two haplotypes designated as hap1 (TGG) and hap2 (GAA) were present. Association analysis between *TaVP1-B* haplotypes and PHS resistance of the 20 wheat cultivars in four experiment environments revealed that the average PHS resistance of accessions with hap1 was significantly better than that of accessions with hap2, which infers the effects of *TaVP1-B* on wheat PHS resistance. To further investigate the impacts of alleles at the *TaVP1-B* locus on PHS resistance, the SNP at 1548 bp of the *TaVP1-B* CDS region was converted to a KASP marker, which was used for genotyping 304 Chinese wheat cultivars, whose PHS resistance was evaluated in three environments. The average sprouting rates (SRs) of 135 wheat cultivars with the hap1 were significantly lower than the 169 cultivars with the hap2, validating the impacts of *TaVP1-B* on PHS resistance in Chinese wheat. The present study provided the breeding-friendly marker for functional variants in the *TaVP1-B* gene, which can be used for genetic improvement of PHS resistance in wheat.

## 1. Introduction

Wheat (*Triticum aestivum* L.) is one of the most important staple food crops, which provides about 20% of the total protein and 21% of the calories for human beings [1,2]. Pre-harvest sprouting (PHS) refers to kernels that germinate on spikes during humid environment or continuous rainfall before harvest [3]. PHS occurs globally in almost all major wheat production areas where it decreases yield and degrades end-use quality and causes about 1 billion dollars loss annually [4,5,6]. In the year 2023, PHS occurred in the Huanghuai wheat production zone in China, including the Henan, Anhui, Hebei, and Jiangsu provinces, and caused serious losses in wheat production [7]. Seed dormancy (SD) is a crucial factor that contributes to PHS, as well as an important trait in domestication of wheat breeding [8]. Other factors such as germination inhibitory substances in chaff tissue, physical barriers to water penetration in the spike, spike structure, etc. may also contribute to PHS resistance [9,10,11,12,13].

PHS is a quantitative trait which is controlled by many QTLs and genes; breeding wheat cultivars with PHS-resistant QTLs/genes is an effective and economic way to decrease the harm of PHS [14]. To date, about 250 QTLs for wheat PHS resistance were detected by QTL mapping and more than 1100 marker-trait association SNP (MTAS) of PHS resistance by GWAS were reported on all 21 chromosomes of wheat [6,11,13,14,15,16,17,18,19,20,21,22,23,24,25,26,27,28,29,30,31,32,33,34,35,36,37,38]. Physical position analysis indicated that many MTAS overlapped with QTLs detected by QTL mapping, and only 29 of the QTLs were major and stable QTLs for PHS resistance over different environments [6,39,40,41].

Currently, genes such as *TaMFT/TaPHS1* [42,43], *TaMKK3-A* [44], *Myb10-D* [5], *TaPI4K-2A* [7], *TaSdr* [45], *TaQsd* [46], *TaDOG1* [47], and *TaVP1* [48] for PHS resistance were cloned by map-based cloning and homology-based cloning. *TaVP1* is a wheat homolog of the *Viviparous-1* (*Vp1*) gene of maize and the *ABSCISIC ACID INSENSITIVE3* (*ABI3*) of *Arabidopsis thaliana* [49,50], which plays an important regulatory role in seed dormancy. Mutants of *vp1/abi3* in maize, rice, and *Arabidopsis* show poor dormancy and are susceptible to PHS [50,51]. Previous studies have demonstrated a positive correlation between the expression level of *TaVP1* and wheat dormancy [48,52]. Copies of *TaVP1* in bread wheat were isolated on the long arm of chromosomes 3A, 3B, and 3D. Although the 3D *TaVP1* was shown to lack sequence variation in modern germplasm, variations in the 3A and 3B copies were associated with PHS susceptibility. The role of this gene is still somewhat poorly understood in bread wheat, but its relationship to PHS was validated by both QTL and association mapping studies, especially *TaVP1*-B [21,30,53]. 

Released wheat reference genomes such as Chinese spring [54,55] and Fielder [56] were effective tools for studying gene structure and function. Recently, genomes of excellent Chinese wheat cultivars Kenong 9204 [57], Aikang58 [58], pan-genome comprised 17 Chinese wheat cultivars [59], and genome of Chinese elite wheat line Zhou8425B [60] were released and uploaded to the website “WheatOmics” (http://wheatomics.sdau.edu.cn/) [61]. In the present study, we analyzed the structure of the *TaVP1-B* gene using the wheat pan-genome (consisting of 20 abovementioned cultivars) and performed an association analysis between *TaVP1-B* and PHS resistance in Chinese wheat cultivars to (1) validate the allelic impacts of *TaVP1-B* on PHS resistance; and (2) develop a high-throughput breeding-friendly marker for *TaVP1-B* that can be used in the genetic improvement of PHS resistance.

## 2. Results

### 2.1. PHS Resistance of Wheat Cultivars

#### 2.1.1. Evaluation of PHS Resistance in 20 Wheat Cultivars with Released Reference Genomes

The PHS resistance of 20 wheat cultivars (Table 1) possessing released reference genomes was evaluated by 3 replicates across four experimental environments, including 3 sowing seasons in Zhoukou and 1 sowing season in Xinyang. The sprouting rates (SRs) of the 20 cultivars varied from 0 to 100% over three years in Zhoukou, and from 5.12% to 100% in 2023 in Xinyang, indicating the presence of a wide range of PHS resistance among these wheat cultivars (Figure 1). Only 2 cultivars, YM158 and Fielder, exhibited SRs lower than 20%, while 12 cultivars had SRs exceeding 50%, suggesting that the majority of modern wheat cultivars possess relatively poor PHS resistance. Correlation analysis revealed that the SRs of these wheat cultivars in the four environments were significantly positively correlated at the 0.01 level, with correlation coefficients ranging from 0.74 to 0.90 (Figure 2). The combined analysis of variance (ANOVA) of 20 cultivars in four environments revealed a significant effect of the genotype (Gen), the environment (Env), and the interaction between genotype and environment (GE_interaction) on SR. The broad-sense heritability was 0.95, indicating that 95% of the phenotypic variation for SR can be attributed to genetic factors (Table 2).

#### 2.1.2. Evaluation of PHS Resistance in 304 Chinese Wheat Cultivars

The PHS resistance of 304 Chinese wheat cultivars was evaluated across three environments and SRs of the accessions ranged from 0 to 100% at all three environments, indicating a substantial amount of variation in PHS resistance in these cultivars. Among the 304 cultivars, only 23 had SRs lower than 10% in all environments (Figure 3 and Appendix A). Correlation analysis between SRs in different environments demonstrated that the SR of the same cultivar had significant positive correlations at the 0.01 level, with correlation coefficients ranging from 0.62 to 0.80 (Figure 4). The combined analysis of variance (ANOVA) of 304 cultivars in three environments revealed a significant effect of the genotype (Gen), the environment (Env), and the interaction between genotype and environment (GE_interaction) on SR, with Gen being the major factor. The broad-sense heritability was 0.87, indicating that 87% of the phenotypic variation for SR can be attributed to genetic factors (Table 3).

### 2.2. Haplotype Analysis of TaVP1-B by Wheat Pan-Genome

The CDS of *TaVP1-B* in 20 wheat cultivars was predicted by the *TaVP1-B* full-length sequences download from released reference genomes. The length of the CDS in each of the 20 cultivars was identical, measuring precisely 1589 bp, which indicated *TaVP1-B* was relatively conserved in these Chinese wheat cultivars. Nevertheless, three SNPs at 496 bp, 524 bp, and 1548 bp of *TaVP1-B* CDS were detected when we checked the alignments of the CDS of *TaVP1-B* and SNP at 496 bp and 524 bp, which were caused Serine/Alanine, Glycine/Asparticacid alterations in the amino acid sequence, respectively (Figure 5). Sequence analysis showed that only two haplotypes were identified in these cultivars and designated as hap1 (TGG) and hap2 (GAA).

### 2.3. Association Analysis Between TaVP1-B Allelic and PHS Resistance in Released Wheat Cultivar Pan-Genomes

To demonstrate whether the alleles of *TaVP1-B* with different haplotypes impact PHS resistance in these 20 wheat cultivars, an association analysis was conducted between SRs of these wheat in four environments and the *TaVP1-B* alleles. Association analysis results showed that the mean SRs of the cultivars with the *TaVP1-B* hap1 allele was significantly lower than those with the hap2 allele in the year 2021–2022 and 2022–2023 at Zhoukou (0.05 level) and 2023–2024 at Zhoukou and Xinyang (0.001 level), which illustrated that *TaVP1-B* alleles impact PHS resistance in these wheat cultivars significantly (Figure 6).

### 2.4. Validation of TaVP1-B Allelic and Effect on PHS Resistance in Chinese Wheat

To further validate the impact on PHS resistance of *TaVP1-B* in Chinese wheat accessions, KASP marker was developed and genotyped in 304 Chinese wheat cultivars or elite breed lines (Table 4).

#### 2.4.1. Haplotypes Identification in Chinese Wheat Accessions

For the conservatism of *TaVP1-B* haplotypes in Chinese wheat and high-throughput genotyping, the G/A SNP at 1548 bp of *TaVP1-B* CDS was transformed to KASP marker and genotyping in 304 Chinese wheat accessions was conducted. The numbers of wheat accessions with *TaVP1-B* hap1 and hap2 were 135 and 169, respectively (Figure 7).

#### 2.4.2. Validation of TaVP1-B in Chinese Wheat Accessions

Association analysis between *TaVP1-B* haplotypes and PHS resistance in Chinese wheat accessions showed that the BLUE of SRs in accessions with *TaVP1-B* hap1 were 39.15%, 40.57%, and 40.36% in the years 2022–2023 and 2023–2024 at Zhoukou and 2023–2024 at Xinyang, respectively, which were significantly lower than those of 54.04%, 52.76%, and 51.74% for accessions with *TaVP1-B* hap2 (Figure 8 and Appendix A). The results above indicated the impact of *TaVP1-B* on PHS resistance in Chinese wheat accessions and that hap1 (TGG) was a favorable haplotype for breeding. A total of thirteen wheat accessions with hap1 were selected for staple PHS resistance (SRs lower than 10% in all environments), which could be used in wheat PHS resistance breeding (Table 5).

#### 2.4.3. Distribution of TaVP1-B Haplotypes in China

The distribution differences in the two *TaVP1-B* haplotypes in Chinese wheat germplasm reveal that the frequency of the hap1 haplotype, which is associated with resistance to pre-harvest sprouting (PHS), is higher in the southern and coastal regions of China (Figure 9). This suggests that hap1 may have played an important role in the natural selection and adaptation processes in these areas. However, compared to regions such as the central, northwestern, southwestern, and northern areas, the hap1 haplotype has not been widely selected or applied in those regions. Consequently, the hap1 haplotype holds great potential for future breeding programs, especially in the genetic improvement of traits related to PHS resistance.

## 3. Discussion

PHS is a globally significant disaster in wheat production, causing losses in both yield and end-use quality with annual economic losses reaching up to USA one billion [9]. In this study, ANOVA and broad-sense heritability analysis suggested that the genetic variation in PHS resistance was mainly controlled by genotype, but the environment and genotype_environment interaction also affected PHS resistance. This consistent finding was reported in a previous study [62]. Environmental such as temperature, light, and humidity in the harvest season of wheat all influenced the PHS [2]; among these, high humidity in the harvest season was the direct cause of PHS in wheat. Hence, in China, PHS occurred frequently in the middle and lower reaches of the Yangtze River wheat region over the past decades because of the recurrent rainfall and humid conditions while it is relatively less likely to occur in other wheat regions. However, with the change in climate, PHS has occurred frequently in the Huanghuai wheat region and the northern wheat region of China in recent years [35]. In 2023, wheat in Henan Province, which contributes over a quarter of China’s total wheat yield, was severely impacted by consecutive rainy days from May 25th to June 2nd. As a direct result, severe PHS occurred during the wheat harvest season, which had a cascading effect on China’s wheat production and economy [7]. According to data from the National Bureau of Statistics of China, China’s wheat yield in 2023 dropped to 134.53 million tons, representing a decrease of 1.226 million tons compared with that in 2022 (https://www.stats.gov.cn/). Due to the severe damage caused by PHS in wheat, how to reduce the losses caused by PHS has always been one of the focal points of wheat breeding. Due to the weather’s unpredictability during the wheat harvest season, it is difficult to accurately avoid rainy days and high-humidity environments. Therefore, identifying QTL or genes conferring PHS resistance and developing breeding-friendly markers for marker-assisted selection are of utmost importance. These tasks can accelerate the breeding, promotion, and planting of PHS-resistant wheat cultivars, which is an effective approach to reduce the damage of PHS in wheat production.

In this study, the PHS resistances of 20 cultivars with reference genomes released were evaluated across four environments. Notably, among these, only the SR of Yangmai158 was lower than 10% in all four environments. In a subsequent investigation, the PHS resistances of 304 Chinese cultivars were assessed in three environments and only 23 cultivars exhibited SRs lower than 10% across all three environments (Appendix A). These findings clearly demonstrate that, in China, the majority of wheat cultivars exhibit poor PHS resistance, likely due to the emphasis on achieving a uniform germination rate and higher yield. In the future process of wheat breeding, more attention should be paid to PHS resistance, and the relationship among yield, germination rate, and PHS resistance should also be balanced.

The *TaVP1-B* gene, located on the 3B chromosome of wheat, shares homology with the Viviparous-1 (*VP-1*) gene of maize and the *ABSCISIC ACID INSENSITIVE3* (*ABI3*) gene of *A. thaliana*. This gene was the focus of prior studies due to its crucial role in conferring resistance to PHS [21,48,53,63,64]. However, the gene structure variants in Chinese wheat accessions remain unclear and few high-throughput diagnostic markers to identify the variants of *TaVP1-B* have been developed, causing low utilization efficiency of this gene in wheat breeding. In this study, the gene structure of *TaVP1-B* was analyzed in 20 cultivars (19 Chinese cultivars included) using data of the released pan-genome, and SNPs and haplotypes of *TaVP1-B* association with PHS resistance were identified. In addition, a high-throughput diagnostic marker for *TaVP1-B* was developed and genotyped in 304 Chinese wheat accessions to validate the allelic impact of *TaVP1-B* and select favorable PHS-resistant accessions. The haplotype distribution of *TaVP1-B* in Chinese wheat cultivars varied significantly across different regions. The hap1 of *TaVP1-B*, which confers resistance to PHS, had a relatively high frequency in the middle and lower Yangtze River wheat region. This might be the result of a long-term natural selection and artificial domestication due to the recurrent rainfall and humid conditions during the wheat harvest season in this region, which may have imposed substantial selective pressure on PHS resistance, leading to the preservation and spread of wheat germplasms with hap1 of *TaVP1-B*. In addition, a total of thirteen Chinese wheat cultivars with hap1 of *TaVP1-B* and staple PHS resistance were identified. The results above could aid us in deciphering the PHS resistance mechanism of *TaVP1-B* and enable more efficient utilization of *TaVP1-B* in wheat breeding. PHS resistance impacted by *TaVP1-B* was validated in our study, but some cultivars with *TaVP1-B* hap2 such as YM158 possess PHS resistance as well, indicating there are other PHS-resistant genes in these accessions, which we could validate in further studies.

With the development of sequencing technologies and bioinformatics, multiple wheat reference genomes were released. However, only reference genomes of 3 Chinese wheat cultivars, “Chinese Spring”, “Kenong 9204”, and “Aikang 58”, have been released, excluding the genomes of 17 wheat cultivars that chronicle the breeding history of China reported in Nature [59]. Following closely, the reference genome of excellent Chinese wheat “Zhou8425B” that applied broadly in breeding was reported [60]. These released Chinese wheat genomes provided favorable data support for studying the genetic mechanisms and gene functions in Chinese wheat. In this study, by using the transgenic recipient wheat “Fielder” and 19 reported Chinese wheat reference genomes (Table 1), allelic variations in the coding region, the impact on PHS in Chinese wheat accessions, favorable haplotypes, and high-throughput diagnostic markers of the *TaVP1-B* were identified and developed. Currently, 46 reference genomes of hexaploid wheat have been released, which can greatly improve the research efficiency of wheat genomics and gene functions.

## 4. Materials and Methods

### 4.1. Plant Materials and Field Trials

A total of 20 wheat cultivars (Table 1) with released reference genomes were used to analyze the structure of the *TaVP1-B* gene. A panel of 304 wheat cultivars from major wheat region of China was used to validate the effect of *TaVP1-B* haplotypes on PHS resistance.

The PHS field trials for the 20 cultivars with reference genomes released were conducted in four environments in the years 2021–2022 (ZK21), 2022–2023 (ZK22), and 2023–2024 (ZK23) at Zhoukou (33°38′24″ N, 114°40′54″ E) and the years 2023–2024 (XY23) at Xinyang (32°17′53″ N, 114°0′25″ E), while PHS resistance of 304 cultivars was evaluated in three environments in 2022–2023 and 2023–2024 at Zhoukou and 2023–2024 at Xinyang. A randomized complete block design (RCBD) with three replicates was used in all environments. Twenty-five seeds were sown in each row, with each plot comprising a 1.5 m row spaced 25.0 cm apart with a general cultivation practice. Spikes of the accessions were harvested at physiological maturity for the evaluation of PHS resistance.

### 4.2. Evaluation of PHS Resistance

The PHS resistance of all wheat cultivars was evaluated by the sprouting rates (SRs) on the spikes. Five spikes of each replicate were harvested from each line at physiological maturity, which is characterized by loss of green color on the spike. Harvested spikes were air dried for 4 days and then stored in a freezer at −20 °C to maintain dormancy. After all lines were harvested, all spikes were taken out from the freezer and air dried again for an additional 2 days. Dried spikes were immersed in de-ionized water for 10 h before they were incubated in a moist chamber at ~23 °C. Wheat spikes were kept at 100% relative humidity in the moist chamber by running a cool humidifier inside the chamber. After 5 days of incubation, quantities of geminated kernels and total kernels of each spike were counted, and the sprouting rates (SRs) were calculated by the percentage of germinated kernels in a spike, which was used as the phenotypic data for PHS resistance analysis.

### 4.3. DNA Isolation of Plant Materials

Leaf tissues sampled at the three-leaf stage and placed in a 2 mL centrifuge tube were freeze dried using a freezing vacuum dryer (Ningbo Scientz Biotechnology Co., Ltd., Ningbo, China) for 24 h, and then ground to fine powder for 2 min at 30 cycles per second in a Mixer Mill (Coyote Bio-science, Beijing, China) with the aid of a 4.0 mm metal bead in each tube. Genomic DNA was isolated by a modified CTAB method [62].

### 4.4. Gene Structure Analysis of TaVP1-B in Wheat Cultivars

The full-length gene sequences of *TaVP1-B* were obtained by “BLAST” and “Get sequence” functions from the website “WheatOmics” (http://wheatomics.sdau.edu.cn/, accessed on 15 December 2024). Coding sequences (CDS) and amino acids sequences of *TaVP1-B* in wheat cultivars were predicted by the tool “ORF finder” on the website “National Center for Biotechnology Information” (https://www.ncbi.nlm.nih.gov/orffinder/, accessed on 15 December 2024). Sequence alignment was conducted by software DNAMAN Version 8.0.

### 4.5. KASP Marker Development and Genotyping 304 Wheat Cultivars

Sequences comprising the targeting SNP found in the gene structure analysis was confirmed by the function “Tools/Get sequence” of “Wheat Omics” and converted to KASP primers by the software “PolyMarker” (https://www.polymarker.info/, accessed on 18 December 2024). Genotyping of the KASP markers in wheat cultivars was carried out in 96-well plates by 10 μL PCR reaction volumes consisting of 5 μL of 2× KASP master mix, 0.14 μL of KASP primer assay mix, and 5 μL of genomic DNA at a concentration of 20 ng μL−1. The PCR protocols were as follows: hot start at 94 °C for 15 min, followed by ten touchdown cycles (94 °C for 20 s; touchdown at 65 °C initially and decreasing by −0.8 °C per cycle for 60 s), followed by 30 additional cycles of denaturation and annealing/extension (94 °C for 10 s; 57 °C for 60 s). Visualization and analysis of the PCR products in 96-well plates were conducted by QuantStudio™ 1 Plus Real-Time PCR System (Thermo Fisher Scientific Inc., Waltham, MA, USA).

### 4.6. Statistical Analysis

The Best Linear Unbiased Estimates (BLUE) of sprouting rate replicates under each environment in the cultivars were calculated using a mixed linear model by genotype as the fixed effect and replicates as the random effects. Multiple comparisons between sprouting rates of accessions with different *TaVP1-B* alleles were calculated by R packages “lme4” and “multcomp”. Figures of sprouting rate distributions and correlations were completed by R packages ”ggplot2” and “corrplot”, respectively. The haplotype distribution map was drawn by the “ggplot2” package in R software (Version 4.3.2) based on the geographical information and haplotypes of the wheat cultivars. Different haplotypes were distinguished by colors and shapes, and the differences in haplotype frequencies were represented by the sizes of the graphics.

The combined analysis of variance (ANOVA) and heritability for sprouting rates of the cultivars in multiple environments were conducted with QTL IciMapping Version 4.2. The linear model used for data analysis was *Y_ijk_* = *μ* + *R_k/j_* + *G_i_* + *E_j_* + *GE_ij_* + *ε_ijk_*_,_ where *Y_ijk_* is the observed value of genotype *i* in replicate k of environment *j*, *μ* is the grand mean, *R_k/j_* is the effect of replicate *k* in environment *j*, *G_i_* is the effect of genotype *i*, *E_j_* is the environment effect, *GE_ij_* is the interaction effect of genotype *i* with environment *j*, and *ε_ijk_* is the error (residual) effect of genotype *i* in replicate *k* of environment *j*. The replicate was treated as a fixed factor, whereas genotype, environment, and their interaction were treated as random factors. Broad-sense heritability on an entry-mean basis was estimated as *H^2^* = *V_G_/*(*V_G_ + V_GE_/e + V_ε_/re*). where *V_G_*, *V_GE_*, and *V_ε_* are the genotypic variance, the variance of genotype_environment interaction, and error variance, respectively. *r* and *e* are the number of replicates and environments, respectively.

## 5. Conclusions

In this study, variants in the CDS region of *TaVP1-B* were detected using the pan-genome approach and genotyped in 304 Chinese wheat accessions, enabling the identification of the allelic impact of *TaVP1-B* on PHS resistance in Chinese wheat accessions. High-throughput diagnostic markers for *TaVP1-B* and excellent PHS-resistant germplasms with the favorable *TaVP1-B* haplotype were selected.

## 6. Patents

The diagnostic KASP marker for *TaVP1-B* mentioned in this study was applied for a national invention patent in China; the patent application number was 2024119713131.1.

## Figures and Tables

**Figure 1 plants-14-00504-f001:**
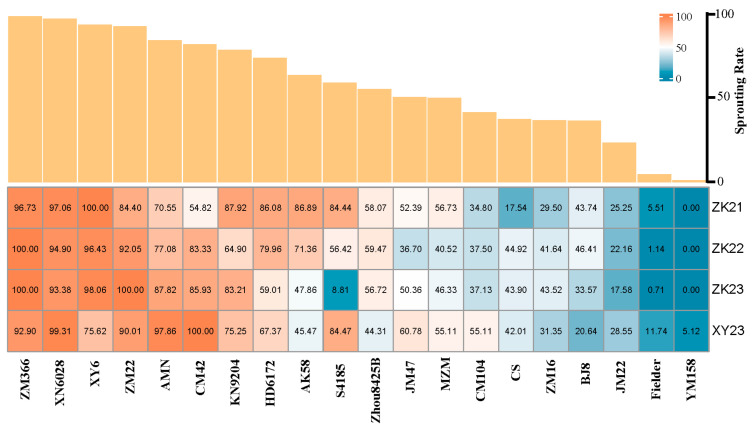
Sprouting rate (%) of 20 wheat cultivars in four environments (ZK21, ZK22, ZK23, and XY23) and mean sprouting rate across environments (shown as vertical bars).

**Figure 2 plants-14-00504-f002:**
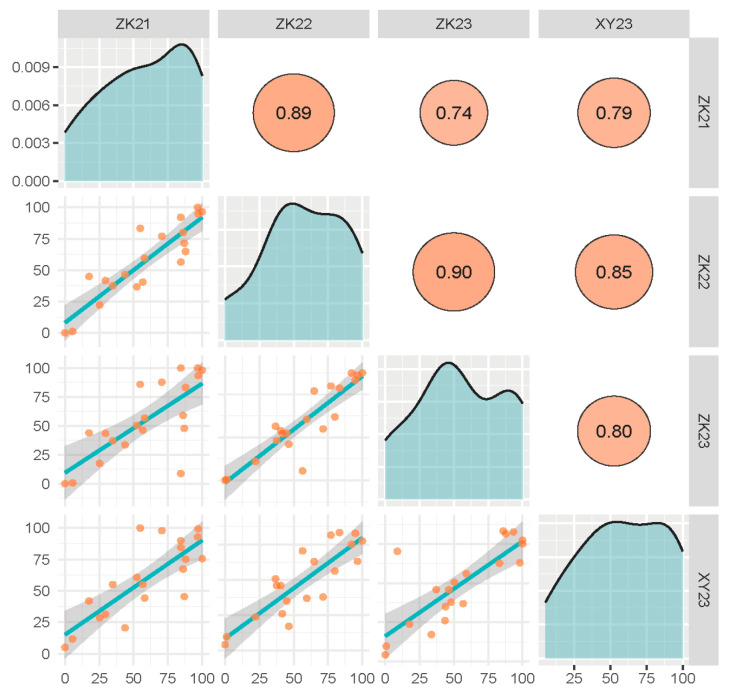
Correlation between environments for sprouting rates of 20 wheat cultivars. Note: The orange dots represent the SRs of the cultivars, and the blue lines are the fitted correlation line.

**Figure 3 plants-14-00504-f003:**
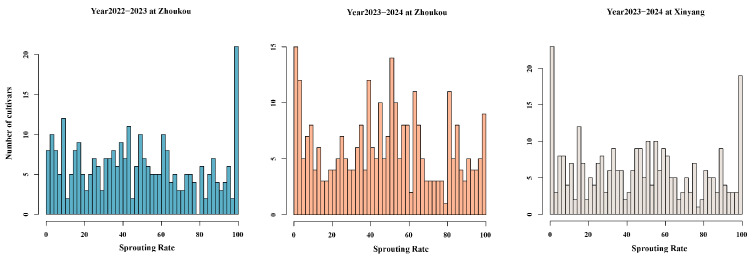
Distribution of sprouting rates (%) of 304 Chinese wheat cultivars in three environments.

**Figure 4 plants-14-00504-f004:**
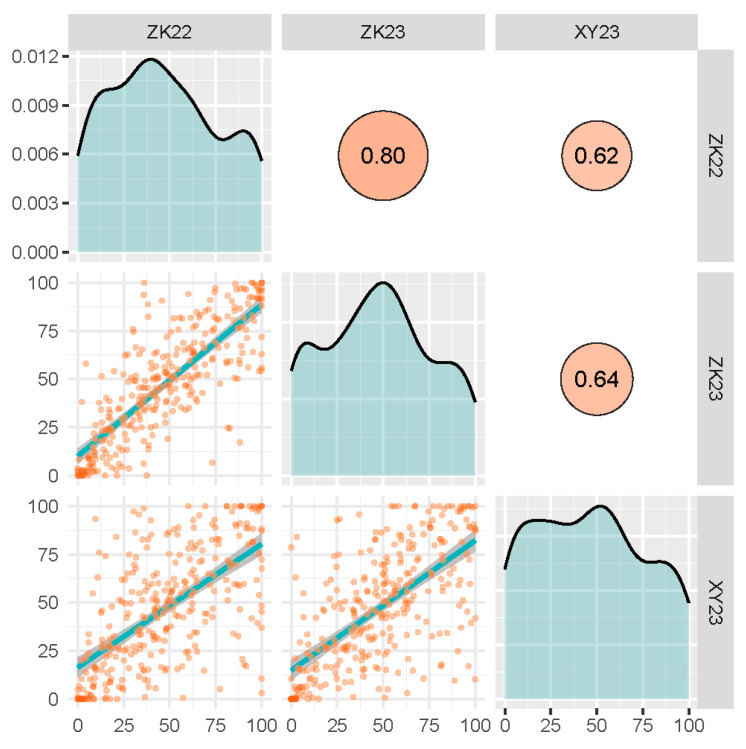
Correlation between environments for sprouting rates of 304 wheat cultivars. Note: The orange dots represent the SRs of the cultivars, and the blue lines are the fitted correlation line.

**Figure 5 plants-14-00504-f005:**
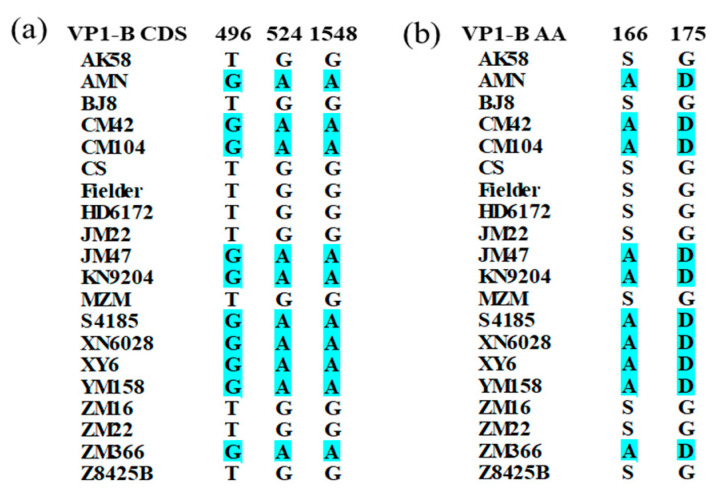
SNPs detected in CDS (**a**) and amino acid variants in protein sequence (**b**) of *TaVP1-B* gene in 20 wheat cultivars.

**Figure 6 plants-14-00504-f006:**
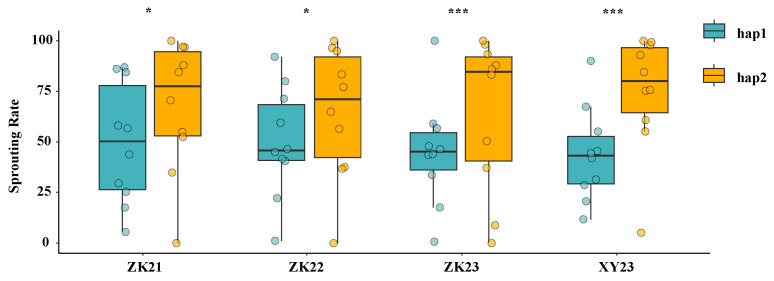
Distribution of sprouting rates in 20 wheat cultivars differing in *TaVP1-B* haplotypes for four environments. Note: * and *** represent significant difference in SRs in wheat with different haplotypes at 0.05 and 0.001 level, respectively.

**Figure 7 plants-14-00504-f007:**
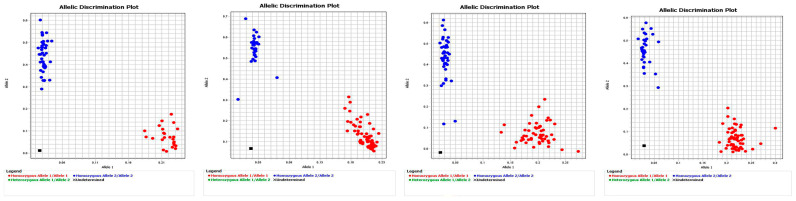
Illustration of KASP assays for *TaVP1-B* haplotype in 304 Chinese wheat cultivars. Note: Red and blue dots represent G (hap1) and A (hap2) labeled alleles, respectively. Black squares represent negative controls (ddH_2_O).

**Figure 8 plants-14-00504-f008:**
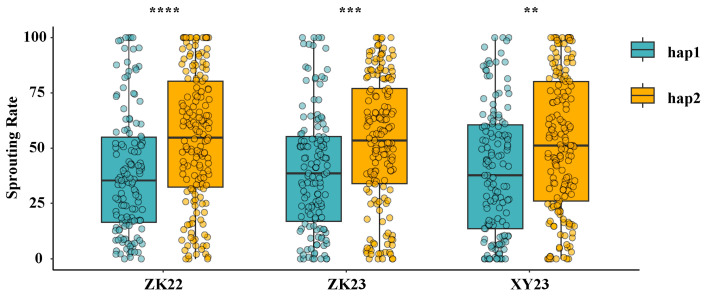
Distribution of sprouting rates in 304 wheat cultivars differing in *TaVP1-B* haplotypes for three environments. Note: **, ***, and **** represent significant difference in SRs in wheat with different haplotypes at 0.01, 0.001, and 0.0001 levels, respectively.

**Figure 9 plants-14-00504-f009:**
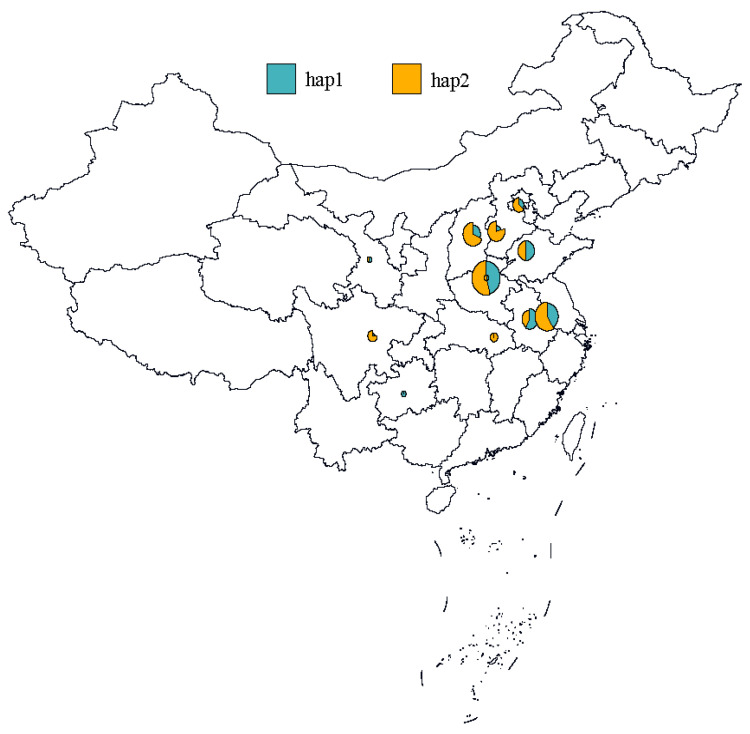
Distribution of *TaVP1-B* haplotypes in China wheat region.

**Table 1 plants-14-00504-t001:** Wheat cultivars and reference genomes released in recent years.

Cultivars	Reference Genomes	Cultivars	Reference Genomes
AK58	doi:10.1016/j.molp.2023.10.015	KN9204	doi:10.1016/j.molp.2022.07.008
AMN	doi:10.1038/s41586-024-08277-0	MZM	doi:10.1038/s41586-024-08277-0
BJ8	doi:10.1038/s41586-024-08277-0	S4185	doi:10.1038/s41586-024-08277-0
CM104	doi:10.1038/s41597-024-03527-2	XN6028	doi:10.1038/s41586-024-08277-0
CM42	doi:10.1038/s41586-024-08277-0	XY6	doi:10.1038/s41586-024-08277-0
CS	doi:10.1126/science.aar7191	YM158	doi:10.1038/s41586-024-08277-0
Fielder	doi:10.1093/dnares/dsab008	Zhou8425B	doi:10.1016/j.xplc.2024.101222
HD6172	doi:10.1038/s41586-024-08277-0	ZM16	doi:10.1038/s41586-024-08277-0
JM22	doi:10.1038/s41586-024-08277-0	ZM22	doi:10.1038/s41586-024-08277-0
JM47	doi:10.1038/s41586-024-08277-0	ZM366	doi:10.1038/s41586-024-08277-0

**Table 2 plants-14-00504-t002:** Analysis of variance for SRs of 20 cultivars in four environments.

Source	DF	SS	MS	*F*-Value	*p*-Value
Gen	19	190,574.97	10,030.26	316.31	0
Env	3	340.96	113.65	3.58	1.53 × 10^−2^
GE_interaction	57	29,690.11	520.88	16.43	0
H^2^ per mean	0.95				

**Table 3 plants-14-00504-t003:** ANOVA and heritability analysis for SRs of 304 cultivars in three environments.

Source	DF	SS	MS	*F*-Value	*p*-Value
Gen	303	1872960.25	6181.39	105.18	0
Env	2	416.57	208.29	3.54	2.91 × 10^−2^
GE_interaction	606	561658.50	926.83	15.77	0
H^2^ per mean	0.87				

**Table 4 plants-14-00504-t004:** KASP marker sequences transformed by SNP at 1548 bp of *TaVP1-B* CDS.

Markers	Sequence (5′-3′)	Note
*TaVP1*-B_CDS1548_HEX	GATGGATGGCAAGAGGTGTTC	hap1 genotype (TGG)
*TaVP1*-B_CDS1548_FAM	GATGGATGGCAAGAGGTGTTT	hap2 genotype (GAA)
*TaVP1*-B_CDS1548_R	CAGGCGTCGGCTTCCAAT	Common marker

**Table 5 plants-14-00504-t005:** Excellent accessions with staple PHS resistance in all environments.

Accessions	Haplotype ^1^	SR_ZK22 (%)	SR_ZK23 (%)	SR_XY23 (%)
Huayu3568	hap1	5.31	9.74	2.68
Lunxuan162	hap1	0.00	1.52	0.00
Neixiang184	hap1	7.06	3.61	2.11
Ningmai16	hap1	3.58	0.72	0.00
Pingan3	hap1	2.57	0.00	0.00
Qimin6	hap1	8.95	2.21	0.00
Shengxuan6	hap1	6.95	6.36	1.56
Shuangji2	hap1	8.40	1.92	6.25
Sui1615	hap1	0.00	0.00	0.00
Taimai1218	hap1	2.93	2.59	4.87
Tianmai159	hap1	5.64	1.75	0.00
Tianmin116	hap1	3.27	0.00	0.00
Zhongxin18	hap1	0.72	8.36	6.57

Note: ^1^ Mean *TaVP1-B* haplotype 1 (TGG).

## Data Availability

The datasets generated and/or analyzed during the current study are available in Wheatomics (http://wheatomics.sdau.edu.cn/).

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
