# Peer review of "Unraveling Allelic Impacts on Pre-Harvest Sprouting Resistance in TaVP1-B of Chinese Wheat Accessions Using Pan-Genome"

_plants, 2025, doi:10.3390/plants14040504_

Round 1

Reviewer 1 Report

Comments and Suggestions for Authors

Dear Authors,

The results presented in your manuscript are interesting for scientific community. However, the manuscript has to be revised to be acceptable for publication.

I gave my specific comments and suggestions for improving the manuscript in the attached file.

Kind regards

Author Response

Dear Reviewer

I hope this letter finds you well. I am writing to express my sincere gratitude for your thorough review and valuable suggestions regarding our manuscript. We have carefully considered each of your comments and have made the necessary revisions to improve the quality of our work.

Comments 1:

Review report Plants-3430693

In the present study, the structure of the gene TaVP1-B, previously reported to be associated with pre-harvest sprouting (PHS) resistance in wheat, was analyzed in 20 wheat cultivars with released reference genomes. The authors identified three SNPs in the coding region of the gene that occurred as only two haplotypes/alleles. Using the identified SNP haplotypes, the authors designed a KASP marker that was validated in a panel of 304 wheat cultivars differing in PHS resistance. The identified alleles (haplotypes) of the TaVP1-B gene were significantly associated with the level of PHS resistance in 20 cultivars with released genomes and in the panel of 304 cultivars.

Previous studies reported allelic variations in the TaVP1-B gene resulting from multiple insertions/deletions in intron regions of the gene. These were used to develop and validate STS and KASP markers in different germplasms. On the other hand, the present study identified a polymorphism in the coding region of the TaVP1-B gene that could be of particular importance for marker-assisted selection for PHS resistance in wheat.

The field experiments performed are well designed and the methods used to identify and validate the SNP polymorphism were appropriate. However, the presentation and interpretation of the results is sometimes not clear enough due to unclear sentences or incorrect statistical interpretation of the results. These include comments on the effects of environment and genotype-environment interaction for sprouting rate in the Results and Discussion section, although no combined analysis of variance across environments was performed for this trait. I therefore suggest that the authors perform ANOVA across environments for both the field trial with 20 cultivars (four environments) and the trial with 304 cultivars (three environments). This is to be expected and was common practice in all earlier multi-environmental studies. The tables and figures are also not clear enough and need to be corrected. A moderate English revision of the manuscript would help to make the sentences clearer. Another major shortcoming in the presentation of the results obtained is that the authors did not provide a list of the 304 Chinese wheat cultivars with the corresponding haplotypes for the TaVP1 gene and the values for the sprouting rate in three environments. These results should be presented as a supplementary table (Table S1) in the revised version of the manuscript.

Response 1:

In the revised version of the article, analysis of variance for SRs of the 20 cultivars and 304 cultivars have been conducted and the results were shown as Table 2 and Table 3. Additionally, details and SRs of all accessions used in this study were list in Table S1.

Specific comments:

Suggestions for improving unclear sentences in certain chapters of the manuscript were made with the help of Track and Changes or by rewriting the sentences

Comments 2:

Abstract:

Lines 23 to 25

TaVP1-B is a homologous gene of the Viviparous-1 (VP-1) gene located on the 3B chromosome of wheat, which has been reported confers to resistance on pre-harvest sprouting (PHS) in wheat several times.”

This sentence suggests that Viviparous-1 (VP-1) is located on the 3B chromosome of wheat, which is incorrect as VP-1 is a maize gene. Therefore, the sentence should be rewritten as follows:

The TaVP1-B gene, located on the 3B chromosome of wheat, is a homolog of the Viviparous-1 (VP-1) gene of maize and has been reported to confer resistance to pre-harvest sprouting (PHS) in wheat.”

Response 2:

We are regarding your comment in lines 23-25 and revised it in the article.

Comments 3:

Lines 25 to 28

„In this study, the structure of the TaVP1-B gene was analyzed using the wheat pan-genome consisting of 20 released cultivars (19 wheat are from China), and three single nucleotide polymorphisms (SNPs) were identified at the 496 bp, 524 bp, and 1548 bp of the TaVP1-B CDS region, respectively

Response 3:

We are regarding your comment in lines 25-28 and revised it in the article.

Comments 4:

Lines 28 to 30

Detected SNPs were not in a high but in complete LD!

Haplotypes analysis showed that these SNPs were in complete linkage disequilibrium and that only two haplotypes, designated as hap1 (TGG) and hap2 (GAA), were present.

Response 4:

We are regarding your comment in lines 28-30 and revised it in the article.

Comments 5:

Lines 34 to 38

The sentence is not clear and there are unnecessary repetitions in the following sentence (“Chinese wheat accessions” and “304 Chinese wheat cultivars”). In addition, the abbreviation (PHS), once introduced, should be used in the rest of the text. “

“To further investigate the impacts of alleles at the TaVP1-B locus on PHS resistance, the SNP at 1548 bp of the TaVP1-B CDS region was converted to a KASP marker, which was used for genotyping 304 Chinese wheat cultivarswhose PHS resistance was evaluated in three environments.”

Response 5:

We are regarding your comment in these lines and revised it in the article.

Comments 6:

Lines 40 to 42

“Functional allele of TaVP1-B and breeding-friendly marker were provided for wheat PHS resistant genetic improving by this study.

The sentence is not clear and I suggest to modify it as follows:

“The present study provided the breeding-friendly marker for functional variants in the TaVP1-B gene which can be used for genetic improvement of PHS resistance in wheat.”

Response 6:

We agree with your suggestions and have made the corresponding revisions.

Comments 7:

  1. Introduction

Lines 68 to 72

“TaVP1 was the homologous gene of Viviparous-1 (Vp1) (homologous to ABSCISIC ACID INSENSITIVE3 of Arabidopsis thaliana, ABI3) in wheat [49,50], which plays a major regulatory role in seed dormancy, with mutants of vp1/abi3 in maize, rice, and Arabidopsis display weak dormancy and are prone to PHS [50,51].”

This sentence is very long and unclear. I suggest modifying it and splitting it into two sentences:

TaVP1 is a wheat homolog of the Viviparous-1 (Vp1) gene of maize and the ABSCISIC ACID INSENSITIVE3 (ABI3) gene of Arabidopsis thaliana [49,50], which plays an important regulatory role in seed dormancy. Mutants of vp1/abi3 in maize, rice and Arabidopsis show poor dormancy and are susceptible to PHS [50,51]

Response 7:

We have revised the sentence according your suggestion.

Comments 8:

Line 83

wheat cultivars [59] and genome of Chinese elite wheat lines Zhou8425B [60] has been

Response 8:

We have revised the “lines” to “line” in the article.

Comments 9: Lines 85 to 91

I have tried to change this section to make it clearer and easier to understand. The first sentence needs to be deleted because it is confusing and unnecessary (the cultivars mentioned above – 20 cultivars - were evaluated in four and not three environments; 304 cultivars were evaluated in three environments!!!)

„In the present study, we analyzed the structure of the TaVP1-B gene using the wheat pan-genome (consisting of 20 abovementioned cultivars) and performed an association analysis between TaVP1-B and PHS resistance in Chinese wheat cultivars to: 1) validate the allelic impacts of TaVP1-B on PHS resistance; 2) develop a high-throughput breeding-friendly marker for TaVP1-B that can be used in genetic improvement of PHS resistance.

Response 9:

We are sorry for our writing error that led to the incorrect mention that the SRs of 20 cultivars were evaluated in three environments. In fact, the SRs of 20 cultivars were evaluated in four environments. We have revised the sentences with your suggestions.

Comments 10:

  1. Results

Line 94

2.1.1. Evaluation of PHS resistance in 20 wheat cultivars with released reference genomes

Response 10:

We followed your suggestion and revised the sentence in the article.

Comments 11:

Lines 97 to 100

“The sprouting rates (SRs) of the 20 cultivars varied from 0 to 100% over three years in Zhoukou, and from 5.12% to 100% in 2023 in Xinyang, indicating the presence of a wide range of PHS resistance among these wheat cultivars (Figure 1)”

Response 11:

We have revised the sentence in the article.

Comments 12:

Lines 106 to 110

However, for certain cultivars, significant differences between different environments were readily observable. For instance, AK 58 and CS had SRs ranging from 45.17% to 86.89% and from 17.54% to 44.92%, respectively, which validates that PHS resistance in wheat is a complex quantitative trait influenced by genetics (G), environment (E), and the interaction between genetics and environment (G×E) (Figure 1).“

This paragraph, which refers to the results shown in Figure 1, is unclearly written and the results obtained were statistically inaccurately interpreted. Since no ANOVA across environments was performed in the present study, there is no information on the significance of the effects of genotype (G), environment (E) and G×E interaction. The authors claim that significant differences were observed between the environments and support this with different ranges of SR values for the two cultivars (AK58 and CS), which is not correct. To determine the significance of the environmental effect, the differences between environmental means (mean of 20 cultivars for each environment) need to be tested based on a combined ANOVA, which was not performed. Similarly, combined ANOVA across environments is needed to determine if the G×E interaction effect is significant.

By comparing SR values in 4 environments for these two cultivars, the authors probably wanted to indicate the presence of a G×E interaction, because cultivar AK58 had twice the SR value in ZK21 (86.89%) than in XY23 (45.47%), while the situation was reversed in CS (17.54% in ZK21 and 42.01% in XY23). This pattern is also obvious when comparing other cultivars, for instance BJ8 and Fielder. However, conclusions about the significance of the G, E and the G×E interaction effects should be based on proper statistical analysis (ANOVA combined over

Response 12:

We have performed ANOVA of SRs in 20 cultivars and analysis the result in lines 104-111 and Table 2.

Comments 13:

Line 116

The caption of Figure 1 is not correct because the figure does not show PHS resistance, but the sprouting rate (%).

“Figure 1. Sprouting rate (%) of 20 wheat cultivars in four environments (ZK21, ZK22, ZK23 and XY23) and mean sprouting rate across environments (shown as vertical bars)”

Response 13:

We have revised the caption of Figure 1 according to your suggestion.

Comments 14:

Line 118

Figure 2. Correlation between environments for sprouting rate of 20 wheat cultivars “

Response 14:

We have revised the caption of Figure 2  as "Correlation between environments for sprouting rates of 20 wheat" cultivars.

Comments 15:

Lines 120 to 123

„The PHS resistance of 304 Chinese wheat cultivars was evaluated across three environments and SRs of accessions ranged from 0 to 100% at all three environments, indicating a substantial amount of variation in PHS resistance in these cultivars.“

Response 15:

We have revised the sentence following your comment.

Comments 16:

Line 123

„Among the 304 cultivars, only 24 had SRs lower than 10% in all environments.“

How is this (24 cultivars) visible from the histogram in Figure 3 with y-axis showing density? The histogram should be revised so that frequency (the number of cultivars) is shown on the y-axis.

Response 16:

We have presented the details and SRs of 304 cultivars (three replicates per year and three years in total) and replotted the Figure 3.

Comments 17:

Lines 124 to 126

„This suggests that in the domestication process of wheat, more attention has been paid to the germination rate during the sowing season for higher yield, while the PHS resistance has been somewhat overlooked (Figure 3).“

This sentence is unclear and does not belong to the Results section. Delete it

Response 17:

Thanks for your suggestion, and we have deleted it in the revised article.

Comments 18:

Lines 126 to 132

The presentation of the correlation results is confusing, as is the labelling of the environments in the corresponding Figure 4. What is the meaning of the labels ZK22_1, ZK22-2, ZK22_3, etc.? Also, the labels on the y-axis for all correlation graphs are between 0 and 100, except for ZK22_1, which ranges from 0.000 to 0.009?

Analogous to Figure 2, where the correlations between four experimental environments are shown, Figure 4 should show the correlations between three experimental environments. Please revise Figure 4 and the comments on the results in the corresponding text.

Response 18:

We have replotted Figure 4 using the average SRs of 304 cultivars in the years 2022-2023, 2023-2024 at Zhoukou (ZK22, ZK23) and 2023-2024 at Xinyang (XY23). Accordingly, the comments on the results were as follows: "Correlation analysis between SRs in different environments demonstrated that the SR of the same cultivar had significant positive correlations at the 0.01 level, with correlation coefficients ranging from 0.62 to 0.80 (Figure 4)."

Comments 19:

Line 134

The caption of Figure 3 is not correct because the figure does not show PHS resistance, but the sprouting rate (%).

Figure 3. Distribution of sprouting rate (%) of 304 Chinese wheat cultivars in three environments”

Response 19:

We have revised the caption of Figure 3 as "Figure 3. Distribution of Sprouting rates (%) of 304 Chinese wheat cultivars in three environments"

Comments 20:

Line 137

Figure 4. Correlation between environments for sprouting rate of 304 wheat cultivars ”

Response 20:

We have revised the caption of Figure 4 as "Figure 4. Correlation between environments for sprouting rates of 304 wheat cultivars"

Comments 21:

Lines 150 to 151

There are unnecessary repetitions in the caption of Figure 5,

“Figure 5. SNPs detected in the CDS (a) and amino acid variants in the protein sequence (b) of the TaVP1-B gene in 20 wheat cultivars.”

Response 21: We have revised the caption of Figure 5 as "Figure 5. SNPs detected in the CDS (a) and amino acid variants in the protein sequence (b) of the TaVP1-B gene in 20 wheat cultivars."

Comments 22:

Line 162

The text in the caption of Figure 6 is inappropriate, as is the labeling of the y-axis in the figure.

Figure 6. Distribution of sprouting rate in 20 wheat cultivars differing in TaVP1-B haplotypes.

The label on the y axis in the figure 6 should be “sprouting rate

Response 22:

We have replotted Figure 6 and revised the label on y axis as "Sprouting Rate". We have revised the caption of Figure 6 to " Distribution of sprouting rates in 20 wheat cultivars differing in TaVP1-B haplotypes for four environments. Note: * and *** represent significant difference of SRs in wheat with different haplotypes at 0.05 and 0.001 level, respectively."

Comments 23:

Lines 183 to 187

In addition to Figure 8 in the text below, please also refer to Table S1. Table S1 (containing the names of 304 Chinese wheat cultivars with the corresponding haplotypes for the TaVP1 gene and the values for the sprouting rate in three environments) should be added as a new table in the supplementary material.

„Association analysis between TaVP1-B haplotypes and PHS resistance in Chinese wheat accessions showed mean SRs of accessions with TaVP1-B hap1 were 39.15%, 40.57% and 40.36% in the year 2022-2023, 2023-2024 at Zhoukou and 2023-2024 at Xinyang, respectively, which were significantly lower than those of 54.04%, 52.76% and 51.74% for accessions with TaVP1-B hap2 (Figure 8 and Table S1).“

Response 23:

We have replotted Figure 8 and the results was revised as "Association analysis between TaVP1-B haplotypes and PHS resistance in Chinese wheat accessions showed that the BLUE of SRs in accessions with TaVP1-B hap1 were 39.15%, 40.57% and 40.36% in the year 2022-2023, 2023-2024 at Zhoukou and 2023-2024 at Xinyang, respectively, which were significantly lower than those of 54.04%, 52.76% and 51.74% for accessions with TaVP1-B hap2 (Figure 8 and Table S1)."

Comments 24:

Line 194

The text in the caption of Figure 8 is inappropriate, as is the labelling of the y-axis in the figure.

„Figure 8. Distribution of sprouting rate in 304 Chinese wheat cultivars differing in TaVP1-B haplotype for three environments.“

The label on the y axis in the figure 8 should be “sprouting rate” and not “PHS”. The explanation of the asterisks should be added in the caption.

Response 24:

We have replotted Figure 8 and revised the caption to "Figure 8. Distribution of sprouting rates in 304 wheat cultivars differing in TaVP1-B haplo-types for three environments. Note: **, *** and **** represent significant difference of SRs in wheat with different haplotypes at 0.01, 0.001 and 0.0001 level, respectively."

Comments 25:

  1. Discussion

Lines 230 to 235

The above text (Lines 230 to 235) should be deleted as its content is repeated in the next paragraph (Lines 241 to 247).

Response 25:

We have deleted the text in the revised article.

Comments 26:

Lines 236 to 238

„TaVP1-B, as a homologous gene of Viviparous-1 (VP-1)gene of maize, is situated on the 3B chromosome of wheat, has been the focus of previous studies due to its putative role in conferring resistance to PHS[21,48,53,62,63].

The sentence is still not clear and should be gramatically improved.

Response 26:

We have revised the sentence as follows: "The TaVP1-B gene, located on the 3B chromosome of wheat, shares homology with the Viviparous-1 (VP-1) gene of maize and the ABSCISIC ACID INSENSITIVE3 (ABI3) gene of A. thaliana. This gene has been the focus of prior studies due to its crucial role in conferring resistance to PHS[21,48,53,62,63]."

Comments 27:

  1. Materials and Methods

Lines 271 to 273

“A total of 20 wheat cultivars (Table 1) with released reference genomes were used to analyze the structure of the TaVP1-B gene. A panel of 304 wheat cultivars from major wheat region of China was used to validate the effect of TaVP1-B haplotypes on PHS resistance .”

Response 27:

We have revised the sentence as follows: "A total of 20 wheat cultivars (Table 1) with released reference genomes were used to analyze the structure of the TaVP1-B gene. A panel of 304 wheat cultivars from major wheat region of China was used to validate the effect of TaVP1-B haplotypes on PHS re-sistance".

Comments 28:

Lines 275 to 281

“The PHS field trials for the 20 cultivars with reference genome released were conducted in four environments, in the years 2021-2022 (ZK21), 2022-2023 (ZK22), 2023-2024 (ZK23) at Zhoukou (33°38′24″N, 114°40′54″E) and year 2023-2024 (XY23) at Xinyang (32°17′53″N, 114°0′25″E), while the PHS resistance of 304 cultivars was evaluated in three environments, in 2022-2023 and 2023-2024 at Zhoukou and 2023-2024 at Xinyang.”

Response 28:

We have revised the sentence as follows: "The PHS field trials for the 20 cultivars with reference genome released were con-ducted in four environments in the years 2021-2022 (ZK21), 2022-2023 (ZK22), 2023-2024 (ZK23) at Zhoukou (33°3824N, 114°4054E) and year 2023-2024 (XY23) at Xinyang (32°1753N, 114°025E), while PHS resistance of 304 cultivars was evaluated in three environments in 2022-2023 and 2023-2024 at Zhoukou and 2023-2024 at Xinyang."

Comments 29:

Line 311

„4.5 KASP marker development and genotyping 304 wheat cultivars

Response 29:

We have revised the sentence in the article as "4.5 KASP marker development and genotyping 304 wheat cultivars"

Response 30:

We have added the mothed of drawing haplotype distribution map in lines 333-337

"The haplotype distribution map was drawn by the “ggplot2” package in R software based on the geographical information and haplotypes of wheat cultivars. Different haplotypes were distinguished by colors and shapes, and the differences in haplotype frequencies were represented by the sizes of the graphics."

Response 31:

We have supplemented the method of ANOVA in the article as lines 338-352

The combined analysis of variance (ANOVA) and heritability for sprouting rates of cultivars in multiple environments were conducted with QTL IciMapping Version 4.2, the replicates were treated as fixed factors whereas genotypes, environment and their interactions were treated as random factors. The linear model used for data analysis was: Yijk = μ + Rk/j + Gi + Ej + GEij + εijk

where Yijk is the observed value of genotype i in replicate k of environment j, μ is the grand mean, Rk/j is the effect of replicate k in environment j, Gi is the effect of genotype i, Ej is the environment effect, GEij is the interaction effect of genotype i with environment j, and εijk is the error (residual) effect of genotype i in replicate k of environment j. Heritability in the broad sense on the base of the mean across replications and environments (or heritability per mean) was estimated by

where VG, VGE, and Vε were variances of genotype, genotype and environment interaction, r and e represent number of replicates and environments, respectively.

Response 32:

We have added the discussion about TaVP1-B haplotype distribution as lines 251-260 in the article.

"The haplotype distribution of TaVP1-B in Chinese wheat cultivars varied significantly across different regions. The hap1 of TaVP1-B, which confers resistance to PHS, had a relatively high frequency in the middle and lower Yangtze River wheat region. This might be the result of a long-term natural selection and artificial domestication due to the recurrent rainfall and humid conditions during the wheat harvest season in this region, which may have imposed substantial selective pressure on PHS resistance, leading to the preservation and spread of wheat germplasms with the hap1 of TaVP1-B. The results above could aid us in deciphering the PHS resistance mechanism of TaVP1-B and enable more efficient utilization of TaVP1-B in wheat breeding."

Response 33:

We have discussed that PHS is significantly influenced by the weather conditions during the harvest season because water is the direct factor causing PHS.

The discussion was in lines 234-244 in revised article:

"PHS is a complex quantitative trait influenced by multiple factors such as genetics, environment and GE reaction[32], among these, water was the direct cause of PHS in wheat, hence in China, PHS occurred frequently in the middle and lower reaches of Yangtze River wheat region over the past decades because of the recurrent rainfall and humid conditions while relatively less likely to occur in other wheat region. However, with the change of climate, PHS occurred frequently in Huanghuai wheat region and Northern wheat region of China in recent years [35]. In 2023, severe PHS occurred in Henan province because wheat in this region was hit by consecutive rainy days from May 25th to June 2nd, which accounts for more than a quarter of China's wheat production, causing significant damage to the country's wheat production and economy [7]."

Thanks again for your review of our article. Happy New Year and hope everything goes well with you!

Yours sincerely,

Wang Danfeng

Reviewer 2 Report

Comments and Suggestions for Authors

Pre-harvest sprouting (PHS) is a trait associated with kernels germination on spikes during humid environment or continuous rainfall before harvest. PHS is an essential characteristic of wheat varieties, especially in monsoon regions such as Japan and China.

PHS is a quantitative trait which is controlled by many QTLs and genes. Currently, genes such as TaMFT/TaPHS1, TaMKK3-A, Myb10-D, TaPI4K-2A, TaSdr, TaQsd, TaDOG1 and TaVP1 for PHS resistance have been cloned by map-based cloning and homology-based cloning. The authors are focusing on TaVP1-B gene among them.

The authors have analyzed the gene structures of TaVP1-B by wheat pan-genome and performed association analysis between TaVP1-B and PHS resistance of Chinese wheat cultivars to: 1) validate allelic impacts of TaVP1-B and its effect on PHS resistance; 2) develop high-throughput breeding-friendly markers for TaVP1-B in PHS resistance genetic improvement.

The objectives of this study are clear and the experiment is well done. The paper needs to be revised in the following MINOR points.

1.       Regarding the evaluation of PHS resistance, can the resistance values of each variety be statistically examined? In other words, which variety is statistically significantly more resistant than any other?

2.       The PHS field test should outline the weather conditions at each location for the seasons 2021-2022, 2022-2023, and 2023-2024. In particular, it is necessary to know what the PHS-related weather conditions were, such as rainfall and solar radiation during the harvest season. We would like a discussion of those weather conditions and the PHS test results.

3.       It is very worthwhile to investigate the haplotype of TaVp1-B in China. It is necessary to discuss even how it relates to the normal climatic conditions in each region.

4.       The figures are all too small to see. They should be made larger or altered to make the figures easier to read.

Author Response

Dear Reviewer

I hope this letter finds you well. I am writing to express my sincere gratitude for your thorough review and valuable suggestions regarding our manuscript. We have carefully considered each of your comments and have made the necessary revisions to improve the quality of our work.

Comments and Suggestions for Authors

1 Pre-harvest sprouting (PHS) is a trait associated with kernels germination on spikes during humid environment or continuous rainfall before harvest. PHS is an essential characteristic of wheat varieties, especially in monsoon regions such as Japan and China.

PHS is a quantitative trait which is controlled by many QTLs and genes. Currently, genes such as TaMFT/TaPHS1, TaMKK3-A, Myb10-D, TaPI4K-2A, TaSdr, TaQsd, TaDOG1 and TaVP1 for PHS resistance have been cloned by map-based cloning and homology-based cloning. The authors are focusing on TaVP1-B gene among them.

The authors have analyzed the gene structures of TaVP1-B by wheat pan-genome and performed association analysis between TaVP1-B and PHS resistance of Chinese wheat cultivars to: 1) validate allelic impacts of TaVP1-B and its effect on PHS resistance; 2) develop high-throughput breeding-friendly markers for TaVP1-B in PHS resistance genetic improvement.The objectives of this study are clear and the experiment is well done. The paper needs to be revised in the following MINOR points.

1.Regarding the evaluation of PHS resistance, can the resistance values of each variety be statistically examined? In other words, which variety is statistically significantly more resistant than any other?

Response:

We have tested the sprouting rates to evaluate the PHS resistance of the wheat. In the revised article, the analyses of variance for sprouting rates of the 20 cultivars and 304 cultivars in this study were performed, the results revealed that genotype, environment and interaction between genotype and environment all had significant impact on wheat PHS resistance. Additionally, we have supplemented a table listing all details and sprouting rates of all cultivars in this paper as Table S1. The cultivars with staple PHS resistance could be selected by the detailed information on sprouting rates in multiple environments.

2.The PHS field test should outline the weather conditions at each location for the seasons 2021-2022, 2022-2023, and 2023-2024. In particular, it is necessary to know what the PHS-related weather conditions were, such as rainfall and solar radiation during the harvest season. We would like a discussion of those weather conditions and the PHS test results.

Response:

During the wheat harvesting seasons in 2021-2022 and 2023-2024, both Zhoukou and Xinyang enjoyed sunny weather. However, in the wheat harvesting season of 2022-2023, Zhoukou was hit by consecutive rainy days from May 25th to June 2nd. Thanks to the city's microclimate, the wheat cultivars planted in Zhoukou reached physiological maturity earlier than those in other regions. As a result, in accordance with the weather forecast (we closely monitor the weather during the wheat harvest season), the spikes of wheat cultivars used for PHS resistance evaluation were harvested before May 23rd. After the consecutive rainy days in 2023, severe PHS occurred in Henan provinces. Regrettably, due to the urgency of harvesting seeds of wheat planted in another places, we failed to obtain the PHS resistance data of the cultivars in the field of Zhoukou.

Nonetheless, we evaluated the PHS resistance by method reported many times in previous studies, these results could also effectively describe the PHS resistance of the wheat cultivars.

Additionally, we have discussed that PHS is significantly influenced by the weather conditions during the harvest season because water is the direct factor causing PHS. The discussion was in lines 234-244 in revised article:

"PHS is a complex quantitative trait influenced by multiple factors such as genetics, environment and GE reaction[32], among these, water was the direct cause of PHS in wheat, hence in China, PHS occurred frequently in the middle and lower reaches of Yangtze River wheat region over the past decades because of the recurrent rainfall and humid conditions while relatively less likely to occur in other wheat region. However, with the change of climate, PHS occurred frequently in Huanghuai wheat region and Northern wheat region of China in recent years [35]. In 2023, severe PHS occurred in Henan province because wheat in this region was hit by consecutive rainy days from May 25th to June 2nd, which accounts for more than a quarter of China's wheat production, causing significant damage to the country's wheat production and economy [7]."

3.It is very worthwhile to investigate the haplotype of TaVp1-B in China. It is necessary to discuss even how it relates to the normal climatic conditions in each region.

Response:

We have discussed the haplotypes distribution of TaVp1-B in China in the lines 251-260 of the revised article.

"The haplotype distribution of TaVP1-B in Chinese wheat cultivars varied significantly across different regions. The hap1 of TaVP1-B, which confers resistance to PHS, had a relatively high frequency in the middle and lower Yangtze River wheat region. This might be the result of a long-term natural selection and artificial domestication due to the recurrent rainfall and humid conditions during the wheat harvest season in this region, which may have imposed substantial selective pressure on PHS resistance, leading to the preservation and spread of wheat germplasms with the hap1 of TaVP1-B. The results above could aid us in deciphering the PHS resistance mechanism of TaVP1-B and enable more efficient utilization of TaVP1-B in wheat breeding."

4.The figures are all too small to see. They should be made larger or altered to make the figures easier to read.

Response:

We have replotted the Figures need to be revised and we will send original figures by compressed package to the editor for the convenience of read and review.

Thanks again for your review of our article. Happy New Year and hope everything goes well with you!

Yours sincerely,

Wang Danfeng

Round 2

Reviewer 1 Report

Comments and Suggestions for Authors

Dear Authors,

In the revised version of the manuscript, you addressed all points from my first review report and corrected the manuscript accordingly. The final version of the manuscript is substantially improved compared to the original version. However, I have identified some additional errors in the newly added text that should be corrected before accepting it for publication.

I have attached one file with my comments and suggestions for corrections („Review report 2_plants 3430693-v2“).

Kind regards

Author Response

Dear Reviewer

I hope this letter finds you well. we sincerely appreciate your patience and the suggestions you provided for our manuscript. We have carefully considered each of your comments and have made the necessary revisions to improve the quality of our work.

Comments 1:

 Results

Lines 121 to 128

The commentary on the combined ANOVA results is confusing and statistically misinterpreted. Genotype and cultivar are synonyms and represent the same effect. Therefore, genotype cannot have a significant effect on PHS resistance of cultivars! Similarly, the interaction between genotype and environment cannot have a significant effect on the PHS resistance of the cultivars, as the interaction itself includes the combined effect of cultivar and environment.

Furthermore, it is neither necessary nor usual to mention all P-values (which can be seen from the table) in the text and to explain that a P-value of 0.029 is smaller than 0.05.

I suggest simplifying the presentation of the ANOVA results as follows:

Analysis of variance (ANOVA) for SRs of the 20 cultivars revealed that P-values of Gen and GE_interaction were both 0.00E+00 (nearly 0), which indicated that the genotype and the interaction between genotype and environment both had extremely significant impacts on PHS resistance of the cultivars. Additionally, P-value of Env was 0.015, which was lower than 0.05, suggesting that the environment had significant effect on the SR of the cultivars as well. The broad-sense heritability was 0.95, indicating that 95% of the phenotypic variation could be attributed to genetic factors (Table 2).The combined analysis of variance (ANOVA) of 20 cultivars in four environments revealed a significant effect of the genotype (Gen), the environment (Env) and the interaction between genotype and environment (GE_interaction) on SR. The broad-sense heritability was 0.95, indicating that 95% of the phenotypic variation for SR can be attributed to genetic factors (Table 2).

Response 1: 

we have revised the "ANOVA" result of 20 cultivars as follows in lines 104-108 in revised manuscript: "The combined analysis of variance (ANOVA) of 20 cultivars in four environments revealed a significant effect of the genotype (Gen), the environment (Env) and the interaction between genotype and environment (GE_interaction) on SR. The broad-sense heritability was 0.95, indicating that 95% of the phenotypic variation for SR can be attributed to genetic factors (Table 2)".

Comments 2:

 Lines 159 to 166

The commentary on the combined ANOVA results is confusing and statistically misinterpreted (explained above, for Lines 121 to 128).

ANOVA for SRs of 304 cultivars revealed that the P-values of both genotype (Gen) and interaction between genotype and environment (GE_interaction) were 0.00E+00 (nearly 0), while P-value of environment (Env) was 0.029, which was less than 0.05. This finding indicated that genotype, environment and interaction between genotype and environment all had significant impacts on PHS resistance of these cultivars. Among them, genotype was the major factor affecting PHS resistance with a broad-sense heritability of 0.87, which suggested that about 87% of the phenotypic variation could be attributed to genetic factors (Table 3). The combined analysis of variance (ANOVA) of 304 cultivars in three environments revealed a significant effect of the genotype (Gen), the environment (Env) and the interaction between genotype and environment (GE_interaction) on SR with Gen being the major factor. The broad-sense heritability was 0.87, indicating that 87% of the phenotypic variation for SR can be attributed to genetic factors (Table 3).

Response 2:

the "ANOVA" of 304 cultivars have been revised follow your suggestion in lines 126-131 in revised manuscript: "The combined analysis of variance (ANOVA) of 304 cultivars in three environments revealed a significant effect of the genotype (Gen), the environment (Env) and the interaction between genotype and environment (GE_interaction) on SR with Gen being the major factor. The broad-sense heritability was 0.87, indicating that 87% of the phenotypic variation for SR can be attributed to genetic factors (Table 3)".

Comments 3: Lines 212 to 218

Why is the content of the table 4 crossed out?

Response 3:

Something wrong here and we added Table 4 again in the revised paper.

Comments 4:

Lines 234 to 236

A total of thirteen wheat accessions with hap1 have beenwere selected for staple PHS resistance (SRs lower than 10% in all environments), which were able tocould be used in wheat PHS resistant resistance breeding (Table 5)

Response 4:

We have revised the sentence according your comments in lines 192-194 in revised manuscript: "A total of thirteen wheat accessions with hap1 were selected for staple PHS resistance (SRs lower than 10% in all environments), which could be used in wheat PHS resistance breeding (Table 5)".

Comments 5:

Materials and Methods

Lines 234 to 236

A randomized complete block design (RCBD) with two three replications replicates was used in all environments.

Response 5:

This sentence has been revised as: "A randomized complete block design (RCBD) with three replicates was used in all environments." in the revised version.

Comments 6:

 Lines 406 to 418

In this paragraph, I suggest moving the explanation of the factor effects (fixed or random) after the description of the model.

I also added the explanation for Vε, which the authors omitted from the text.

„The combined analysis of variance (ANOVA) and heritability for sprouting rates of cultivars in multiple environments were conducted with QTL IciMapping Version 4.2. the replicates were treated as fixed factors whereas genotypes, environment and their interactions were treated as random factors. The linear model used for data analysis was: Yijk = μ + Rk/j + Gi + Ej + GEij + εijk where Yijk is the observed value of genotype i in replicate k of environment j, μ is the grand mean, Rk/j is the effect of replicate k in environment j, Gi is the effect of genotype i, Ej is the environment effect, GEij is the interaction effect of genotype i with environment j, and εijk is the error (residual) effect of genotype i in replicate k of environment j. Replicate was treated as a fixed factor, whereas genotype, environment and their interaction were treated as random factors. Broad-sense hHeritabilty in the broad sense on the base of the mean across replications and environments (or heritability per mean)on an entry-mean basis was estimated by as: H2=VG/(VG+VGE/e+Vε/re),

where VG, VGE, and Vε were are the genotypic variances of genotype, the variance of genotype- and environment interaction, and error variance, respectively and r and e represent are the number of replicates and environments, respectively.“

Response 6:

The paragraph has been revised as: "The combined analysis of variance (ANOVA) and heritability for sprouting rates of cultivars in multiple environments were conducted with QTL IciMapping Version 4.2.  The linear model used for data analysis was: Yijk = μ + Rk/j + Gi + Ej + GEij + εijk, where Yijk is the observed value of genotype i in replicate k of environment j, μ is the grand mean, Rk/j is the effect of replicate k in environment j, Gi is the effect of genotype i, Ej is the environment effect, GEij is the interaction effect of genotype i with environment j, and εijk is the error (residual) effect of genotype i in replicate k of environment j. Replicate was treated as a fixed factor, whereas genotype, environment and their interaction were treated as random factors. Broad-sense heritability on an entry-mean basis was estimated as: H2=VG/(VG+VGE/e+Vε/re). where VG, VGE, and Vε are the genotypic variance, the variance of genotype_environment interaction, and error variance, respectively. r and e are the number of replicates and environments, respectively." in lines 343-354 in revised manuscript according to your suggestion.

Response 7: line 122-123

Due to an oversight where we mistakenly counted the table header as a cultivar, we have now revised the number of wheat cultivars with sprouting rates (SRs) below 10% across three experimental environments to 23.

"Among the 304 cultivars, only 23 had SRs lower than 10% in all environments (Figure 3 and Table S1)."

Response 8:

We have reworked the first paragraph of the discussion, spanning lines 224-250, to enhance its logical flow and incorporate more specific data for robust support.

"PHS is a globally significant disaster in wheat production, causing losses in both yield and end-use quality with annual economic losses reaching up to one billion U.S. dollars [9]. In this study, ANOVA and broad-sense heritability analysis suggested that the genetic variation of PHS resistance was mainly controlled by genotype, but the environment and genotype_environment interaction also affected PHS resistance. The consistent finding was reported in the previous study [64]. Environmental factors such as temperature, light and humidity in the harvest season of wheat all influenced the PHS [2], among these, high humidity in the harvest season was the direct cause of PHS in wheat. Hence in China, PHS occurred frequently in the middle and lower reaches of Yangtze River wheat region over the past decades because of the recurrent rainfall and humid conditions while it is relatively less likely to occur in other wheat region. However, with the change of cli-mate, PHS has occurred frequently in Huanghuai wheat region and Northern wheat region of China in recent years [35]. In 2023, wheat in Henan province, which contributes over a quarter of China's total wheat yield, was severely impacted by consecutive rainy days from May 25th to June 2nd. As a direct result, severe PHS occurred during the wheat harvest season, which had a cascading effect on China's wheat production and economy [7]. According to data from the National Bureau of Statistics of China, China's wheat yield in 2023 dropped to 134.53 million tons, representing a decrease of 1.226 million tons compared with that in 2022 (https://www.stats.gov.cn/). Due to the severe damage caused by PHS in wheat, how to reduce the losses caused by PHS has always been one of the focal points of wheat breeding. Due to the weather's unpredictability during the wheat harvest season, it is difficult to accurately avoid rainy days and high-humidity environments. Therefore, identifying QTL or genes conferring PHS resistance and developing breeding-friendly markers for marker-assisted selection are of utmost importance. These tasks can accelerate the breeding, promotion and planting of PHS resistant wheat cultivars, which is an effective approach to reduce the damage of PHS in wheat production.

 Response 9: we have discussed the PHS resistance of Chinese wheat in lines 251-260 in revised paper:

" In this study, the PHS resistances of 20 cultivars with reference genomes released were evaluated across four environments. Notably, among these, only the SR of Yangmai158 was lower than 10% in all four environments. In a subsequent investigation, the PHS resistances of 304 Chinese cultivars were assessed in three environments and only 23 cultivars exhibited SRs lower than 10% across all three environments (Table S1). These findings clearly demonstrate that, in China, the majority of wheat cultivars exhibit poor PHS resistance, likely due to the emphasis on achieving a uniform germination rate and higher yield. In the future process of wheat breeding, more attention should be paid to PHS resistance, and the relationship among yield, germination rate and PHS resistance should also be balanced."

Thank you again for your suggestions on our submission. I wish you all the best and Happy Chinese New Year!

Best regards,

Danfeng

2025.02.03
